# Prevalence, Symptoms, and Triggering Factors of Panic Attacks among Dental Students in Riyadh Saudi Arabia—A Cross Sectional Survey

**DOI:** 10.3390/healthcare11222971

**Published:** 2023-11-16

**Authors:** Sanjeev B. Khanagar, Reema Jamal Altuwayjiri, Nadeen Mohammed Albarqy, Ghida Ahmed Alzahrani, Hibah Ali Alhusayni, Sarah Yousef Alsaif

**Affiliations:** 1Preventive Dental Science Department, College of Dentistry, King Saud bin Abdulaziz University for Health Sciences, Riyadh 11426, Saudi Arabia; 2King Abdullah International Medical Research Centre, Ministry of National Guard Health Affairs, Riyadh 11481, Saudi Arabia; 3College of Dentistry, King Saud bin Abdulaziz University for Health Sciences, Riyadh 11426, Saudi Arabia; altuwayjiri257@ksau-hs.edu.sa (R.J.A.);

**Keywords:** dental students, impact, panic attacks, response, symptoms, triggering factors

## Abstract

Panic disorder by definition is an anxiety disorder of unexpected and repeated episodes of intense fear. Panic attacks are usually diagnosed by four or more of a set of symptoms that include palpitations, sweating, trembling, shortness of breath, chest pain, nausea, dizziness, and hot flushes. They usually interfere with daily life situations and also interfere with education. Hence, the aim of this study was to assess the prevalence of panic attacks, their symptoms, and triggering factors among dental students in Riyadh, Saudi Arabia. Data were collected from 394 students using a structured and validated questionnaire. The prevalence of panic attacks among dental students in Riyadh, Saudi Arabia, was 42.9%. Most of the participants who experienced higher episodes of panic attacks were females (53.4%) when compared to males (24.5%). Third year students displayed greater (58.3%) episodes of panic attacks compared to their respective counterparts. The most reported symptom of panic attacks was rapid or pounding heartbeat followed by breathlessness, chest pain, and shaking or trembling. It was also noted that most of the participants (63.31%) encountered a panic attack for the first time after joining dental school. The situations where dental students frequently experienced panic attacks were during exams, clinic procedures, giving presentations, and especially while under a lot of stress. The high occurrence of panic attacks among dental students highlights the importance of providing support programs and implementing preventive measures to help students, particularly those who are most susceptible to higher levels of these psychological conditions. Dental students experiencing panic attacks should be provided with necessary counseling sessions or psychiatric consultation in order to overcome such scenarios. Dental schools should consider these findings when planning the dental curriculum. Hence, the role of the faculty members is essential in these situations to provide support for the affected students.

## 1. Introduction

According to the National Institute of Mental Health, panic disorder is defined as an anxiety disorder of unexpected and repeated episodes of intense fear. Panic attacks, according to the National Comorbidity Survey-Replication (NCS-R), have a prevalence of 4.7% among the general population throughout a person’s lifetime [1]. High levels of social, marital, occupational, and physical disability result from these attacks [2]. They are one of the costliest mental health conditions in the community and in primary healthcare [1].

Panic attacks are diagnosed by four or more of a set of symptoms that include palpitations, sweating, trembling, shortness of breath, chest pain, nausea, dizziness, and hot flushes [3]. Studies have reported that the most severe symptoms experienced were palpitations, sweating, and trembling, while the most common symptoms experienced were palpitations, trembling, sweating, and shortness of breath [4,5,6]. Patients with undiagnosed panic disorder may experience these symptoms and episodes if they are unaware of their condition [7].

The occurrence of panic attacks can vary from several times per day to only a few attacks per year. Even though these attacks are a key feature of panic disorder, panic attacks can accompany mental and even medical disorders [1]. Even though panic attacks only last a few minutes, they have a major influence on the patients’ quality of life [8]. They greatly interfere with daily life situations, especially with education. Their occurrence may cause issues with students’ academic performance, including missing workdays and avoiding situations that may trigger a panic attack [9]. Suicide risk is elevated in individuals suffering from panic disorder because they cannot manage their social and family life normally, and it may be linked to a deterioration in quality of life [7]. The disorder is also linked to a higher incidence of smoking and co-morbid medical conditions [7].

Evidence of direct inheritance of the disorder from one generation to the next has been found [8]. For instance, the condition was seen in 15–17% of relatives of individuals with panic disorder, while mono chorionic twin concordance was 85–90% and heritability was 48% [8]. Two intricate protective genotypes associated with panic disorder have been identified. This can be due to a predisposition to the disease associated with their genes [8]. According to recent research, some individuals are predisposed to the disorder, which typically starts in young adulthood and affects women twice as often as men, with a low prevalence in children under the age of 14 [8]. Panic disorder peaks in adolescence and early adulthood [9].

Several studies have reported on the prevalence of panic attacks among college students [10,11]. Ramachandran et al. [10] reported an increase in panic attacks experienced by medical students. The study concluded that half of them suffer from mild panic disorder, while one third suffer from moderate panic disorder. Nazemi et al. [11] reported on panic attacks among university students in Iran. Thirty-eight percent of participants reported at least one panic attack in the past year and 21.4% reported at least one panic attack in the past 4 weeks [11].

A study conducted by Abbasi et al. [12] comparing stress level between dental and medical students reported slightly higher stress (96.7%) prevalence experienced among dental students. According to previous studies, dental school is more stressful than medical school, this variation has been linked to the more advanced and refined psychomotor abilities demanded of dental students in dental education [13,14].

Several studies have been conducted among dental students discussing stress and anxiety [15,16,17,18,19]. Studies have reported that dental students experience high stress during their time in dental school for a variety of reasons, including demanding curricula, meeting course and clinical requirements, managing anxious patients, having limited time, exams, and having higher expectations for their academic performance [20,21,22,23]. Studies have reported that the transition from preclinical to clinical years for students is more stressful due to the expectation of academic achievement and other requirements in dentistry [24,25]. Stress can have a negative impact on dental students’ physical and mental health, which pose a threat to their overall wellbeing [26]. Increased stress among dental students eventually leads to psychological issues and emotional distress, which causes professional burnout and decreased productivity [27].

The higher prevalence of stress level among dental students suggests the possibility of experiencing panic attacks throughout their academic years. In addition, multiple studies have reported on panic attacks occurrence in university students [10,11]. However, there were no studies reporting the prevalence of panic attacks specifically among dental students. Hence, there is a need to know the prevalence of panic attacks among dental students and its impact on their academic and clinical performance. These findings can help dental students identify the causes of these panic attacks and manage them accordingly. Therefore, the aim of the study was to know the prevalence of panic attacks among dental students, their symptoms, and triggering factors. The objective of the study is to compare the results between genders and academic years and universities, and to know the impact of these panic attacks on students’ academic and clinical performance.

## 2. Materials and Methods

### 2.1. Study Design

This is a cross-sectional analytical study. Prior to the initiation of the research, ethical clearance was obtained from the Institutional Review Board, King Abdullah International Medical Research Center, Riyadh, Saudi Arabia (KAIMRC). (No. IRB/1623/22).

### 2.2. Sample Size and Sampling Frame

A pilot study was conducted in order to estimate the sample size for the main study. Based on the findings, with an expected proportion of 79.6% from the pilot study and relative precision of 5% and confidence level of 95%, the calculated sample size came up to 394. There are 6 dental colleges in Riyadh, Saudi Arabia, and all these colleges were included in this study. Three of these colleges are governmental, while the remaining three are private colleges. The governmental colleges included are King Saud bin Abdulaziz University for Health Sciences, King Saud University, and Princess Nourah bint Abdulrahman University. While the private colleges are Riyadh Elm University, Dar Al Uloom University, and Vision college. A simple random sampling technique was considered for selecting a minimum of 70 participants to fulfil the quota from each university upon fulfilling the eligibility criteria. However, during the sampling process, only those participants present on the day of sample selection at the institutions were considered; hence, this may have contributed to the sampling error.

### 2.3. Eligibility Criteria

Inclusion criteria: dental students studying in their clinical academic year and those agreeing to participate by signing the informed consent form.

Exclusion criteria: dental students suffering from mental illnesses or suffering from psychiatric issues.

### 2.4. Data Collection

Data for this study were collected using a structured questionnaire in the English language, which was developed after referring to similar study [11]. It was available both as a personal hard copy and an online Web-based questionnaire. It comprised 17 questions divided into six sections. The first section included three questions (Q1–Q3) related to the participants’ demographic data. The second section included one screening question, assessing the participant’s experience of a panic attack through a set of panic attack symptoms (Q4). The third section comprised five questions (Q5–Q9) assessing the severity and frequency of panic attacks. The fourth section comprised three questions (Q10–Q12) related to the triggering factors of the panic attacks. The fifth section comprised two questions (Q13–Q14) related to the impact of panic attacks on their performance. The last section comprised three questions (Q15–Q17) related to participants’ response towards panic attacks. The data collected from participants were further entered and analyzed using the SPSS statistical program, Version 29 (IBM Corporation, Armonk, NY, USA).

### 2.5. Validity and Reliability of Questionnaire

In order to assess the content validity of the questionnaire, the questionnaire was distributed among a panel of 6 content experts. The panel members had good research experience. The purpose was to assess the degree of agreement between the members for the questions included, which was assessed using Aiken’s V test. In this questionnaire, a value greater than 0.90 was achieved for each question.

And in order to assess the reliability of this tool, a pilot study was conducted on 5% of the study samples from the same institution. To assess the test–retest reliability, data were collected from the same study participants after a two week gap. An intra-class correlation coefficient value of 8 was obtained, which indicated good reliability of the tool. Hence, no further changes were made in the final version of the questionnaire. The responses obtained from the pilot study were included in the final data for analysis.

### 2.6. Study Participants

Data were collected from the study participants from their respective institutions after obtaining data collection permission from the institutions’ authorities. Personal hard copy was distributed and a web-based electronic survey using Google Forms was also used. The data required for this study were gathered for a period of two months (from 3 December 2022 to 1 February 2023).

### 2.7. Statistical Analysis

The data analysis was performed using the SPSS software, Version 29 (IBM Corporation, Armonk, NY, USA). Descriptive statistics were calculated and Chi square analysis and Multiple Logistic Regression Analysis were used to measure the association of gender and university on their responses. Statistical significance was set at *p* ≤ 0.05.

## 3. Results

### 3.1. Demographic Details of Study Participants

In the present study, a total of 580 students were approached to participate in the research. However, 394 responded; hence, the response rate was 67.93%. The demographic data revealed that 251 (63.71%) study participants were females and 143 (36.29%) were male participants. In the present study, 102 (25.89%) of the study participants were in their fifth academic year. In the present study, 50 (12.69%) of the study participants were from Dar Al Uloom University, 105 (26.65%) were from King Saud Bin Abdulaziz University for Health Sciences, 106 (26.90%) were from King Saud University, 64 (16.24%) were from Princess Nourah Bint Abdulrahman University, 64 (16.24%) were from Riyadh Elm University, and 5 (1.27%) were from Vision College (Table 1).

### 3.2. Prevalence and Symptoms of Panic Attacks among Study Participants

A total of 169 (42.9%) participants reported having experienced panic attack symptoms. Out of these, 49 (28.99%) of them stated that they encountered panic attacks during the middle of the academic year, while 60 (35.50%) of them could not recall when it had occurred. More than half of the study participants, 99 (58.58%), reported that they had experienced 1–5 episodes of panic attacks in the past four weeks. In terms of the panic duration, the majority, 93 (55.03%), of them had panic attacks that lasted for less than ten minutes and a small number of students, 7 (4.14%), experienced them for more than one hour. In addition, 64 (37.87%) of them had mild anxiety of having another attack in the future. A total of 62 (36.69%) of them mentioned that rapid or pounding heartbeat was the most common symptom experienced during panic attack, whereas 26 (15.38%) had breathlessness, 15 (8.88%) felt chest pain, and 10 (5.92%) had shaking or trembling during the period of the panic attack. Most of the participants, 107 (63.31%), agreed that the first time they encountered a panic attack was after joining dental school, as shown in Table 2.

### 3.3. Triggering Factors of Panic Attacks

More than half of the participants, 88 (52.07%), reported that they experienced panic attacks mostly before or during exams, in the clinic while performing procedures, while giving a presentation, and especially while under a lot of stress. About 9 (5.3%) of them revealed that attacks occurred unexpectedly and “out of the blue”. The majority, 115 (68.05%), of the participants reported that they experienced panic attacks outside of the college, as shown in Table 3.

### 3.4. Impact on Performance among Study Participants

A total of 80 (47.34%) study participants believed that panic attacks had a mild impact on their academic/clinical performance, and most of them, 143 (84.62%), consider that they are harmful for their physical or mental health, as shown in Table 4.

### 3.5. Response of Panic Attacks among Study Participants

The majority of the participants, 89 (52.66%), did not show any fear or avoidance in going to places such as clinics, classrooms/exam halls, while 34.32% stated that they had mild or occasional fear after experiencing a panic attack. When asked were there any activities that they avoided as they feel it may trigger panic attack, 72 (42.60%) of them had no fear/avoidance; 63 (37.28%) reported mild occasional fear/avoidance; and 27 (15.98%) stated moderate noticeable fear/avoidance. A total of 44 (26.04%) study participants reported that they had consulted a counsellor to manage their panic attacks, as shown in Table 5.

### 3.6. Association between Occurrence of Panic Attacks, Gender, Academic Year, and University Enrolled

A Chi square analysis revealed that most of the participants who experienced higher episodes of panic attack were females (53.4%) when compared to males (24.5%), with a *p* value of <0.001. Out of 48 students in their third year, a greater majority (58.3%) had reported to have panic attack compared to their respective counterparts and this difference was found to be statistically significant (*p* value = 0.01). Most of the students (59.4%) enrolled in Princess Nourah Bint Abdulrahman University encountered greater rates of panic attack, with a statistical significance of *p* value of <0.001, as shown in Table 6.

## 4. Discussion

The field of dentistry is regarded as one of the most stressful health professions for a variety of reasons, including the unavoidable pressure to perform well academically, high competition with other students, exam anxiety, meeting academic and clinical requirements, and worrying about career choices [21,28,29,30,31,32,33,34,35,36,37,38,39].

In the present study, 42.9% of dental students reported having experienced panic attacks. The prevalence of panic attacks among dental students showed similarities with results of other studies reported in the literature [6,11]. Norton et al. [6] reported that 22.1% of undergraduate students at the University of Houston experienced panic attacks. In addition, Nazemi et al. [11] reported a prevalence of 38% among university students in Iran. In the present study, 28% of dental students reported experiencing these attacks in the middle of the year, which could be related to the increasing stress or workload on students at that time. Studies have also reported other factors responsible for the increased amount of stress among dental students, which include time management and patient scheduling pressures, managing uncooperative patients, and the highly technical and intensive nature of work [40]. On the other hand, 35% of the study participants could not recall whether they occurred at the beginning, middle, or end of the year. However, previous studies discussing panic attacks have not reported their occurrence time in the duration of the academic year [9,10,11,33]. In the present study, 58% of the study participants reported experiencing one to five panic attacks in the past four weeks. These findings were higher than the study reported by Nazemi et al. [11] on college students, where only 21.4% reported experiencing panic attacks. This indicates that dental students experienced panic attacks more frequently than other professional students. The higher frequency of panic attacks among dental students could be due to the higher amount of stress they undergo in comparison to other professional students [20,40].

In the present study, 55.03% of the study participants reported that the duration of the panic attacks was less than ten minutes, which is similar to the mean time of an average panic attack, eight minutes, as reported in a study conducted by Meuret AE et al. [31]. In this study, 4.14% of the study participants reported experiencing panic attacks for more than one hour. A study by Nazemi et al. [11] reported that 43.2% of their study participants had experienced panic attacks that lasted less than 10 min, and 14.4% experienced attacks that lasted for a couple of hours. In the present study, various symptoms were reported along with these attacks, which included rapid or pounding heartbeat (36.69%), breathlessness (15.38%), and chest pain (8.88%). Norton et al. [6] reported similar experienced symptoms among undergraduate students, which included heart pounding (74%), rapid heartbeats (67.1%), and dizziness (66.2%), while another study reported palpitations, sweating, and trembling among the most severely rated symptoms by participants [11].

In the present study, there was a significant difference noted for academic year and the prevalence of panic attacks. These findings were similar to the results of the study reported by Naidu R.S et al. [41], where they found higher stress levels among dental students as they progress in the academic program, especially during their transition into clinical years. Several studies have reported the factors associated with the increased amount of stress with academic progress, which includes stress related to future uncertainties about their carrier in dentistry, fear of unemployment after graduation, and fear of reduced possibilities of pursuing higher education [20,42]. Carrier uncertainty among dental graduates is a global concern; hence, the same is noted in Saudi Arabia as there has been a drastic increase in the number of dental colleges in the past few decades which has resulted in increasing number of dental graduates every year.

In the present study, 47.34% believed that experiencing a panic attack had a slight impact on their academic or clinical performance. However, Mark E et al. [9] reported cases of panic disorder in university students who felt destined to fail academically when they experienced panic attacks. Additionally, Rapee et al. [32] reported the impact of life events on subjects with panic disorder. The findings indicate that the thought patterns of people during panic attacks are characterized by fears of immediate drastic outcomes such as death, insanity, or loss of control. Such patterns obviously affect students’ ability to function both socially and academically.

In the present study, the majority of the participants, specifically 84.62%, believed that panic attacks can have negative effects on their physical or mental health. Similarly, other studies have demonstrated that panic attacks can have a significant impact on one’s quality of life and result in depression and disability [7,33]. Moreover, panic attacks can lead to a reduced quality of life as patients may struggle to function normally in their social and family life [7]. According to Michelson et al. [34] panic disorders tend to worsen in severity, frequency, and duration if left untreated and can further result in hypochondriasis, chronic anxiety, and phobic avoidance.

In the current study, the majority of participants (52.66%) did not display any signs of fear or avoidance when it came to attending places like clinics, classrooms, or exam halls; however, 34.32% reported experiencing mild or occasional fear after experiencing a panic attack. On the other hand, a study conducted by Nazemi et al. [11] among university students in Iran found that those students who had experienced panic attacks reported greater avoidance of relaxation states and exams. These findings align with previous research suggesting that states of relaxation can suddenly reverse and lead to panic [35].

Regarding activities that participants avoided due to the possibility of triggering a panic attack, 42.60% did not have any fear or avoidance, while only 4.14% displayed severe and extensive avoidance. Additionally, 26.04% of study participants confirmed seeking the help of a counselor to manage their panic attacks. On the contrary, the study of Nazemi et al. [11] in Iran found that the majority of participants (52.3%) tried to forget about their panic attacks, while 24.2% reported talking to friends or family members about their experience and some participants sought medical (4.5%) or psychological (4.5%) treatment as well as counseling (7.6%). Because of the unexpected nature of panic attacks and the lack of known triggering factors, they are very difficult to anticipate, leaving those who encounter them feeling unable to control themselves. In this study, the majority of participants, 68.05%, stated that they had panic episodes both inside and outside of the college. In total, 52.07% of the study participants have identifiable triggers for their attacks, while 5.3% stated that attacks had happened suddenly and out of the blue. Similarly, Ellen W et al. [36] reported that the majority of participants (95%) were able to identify a trigger for their attack.

Several epidemiological studies have reported on gender variations in panic attacks. Jyoce P.R. et al. [43] reported that the lifetime prevalence of panic attacks is higher among females in comparison with males. Katerndahl D.A. et al. [44] also reported that the lifetime prevalence of panic states is higher among females in comparison with males; however, the findings were not statistically significant. Another study conducted by Reed V. et al. [45] also reported that the prevalence and onset of panic disorders is higher among females in comparison with males. In the present study, female participants (53.4%) experienced more episodes of panic attacks when compared with males (24.5%). Similarly, previous studies reported higher levels of depression, anxiety, and stress between females when compared with male students [37,38,39]. This could be because females tend to have higher levels of neuroticism than males, which can lead to hormonal changes that could more significantly affect their emotions [46]. According to several studies, females in Saudi Arabia have displayed higher level of stress [21,39,47,48,49]. On the other hand, only one study reported that male students experienced higher levels of distress when compared to female students [17]. Another study reported similar levels of stress among male and female students [19]. Regarding the relation between university enrolment and the likelihood of panic attacks, Princess Noura Bint Abdulrahman University was associated with higher rates of panic attacks in comparison with other dental universities; this could be because this university is exclusively for female students.

Considering the frightening and distressing nature of panic attacks, it is important to understand the measures required for preventing and managing these panic attacks. It is important to practice appropriate breathing exercises to calm and prevent panic attacks [50,51]. Practicing regular physical exercises like aerobic exercises can help in managing stress levels, improving the individuals’ mood and boosting the confidence. It is also important to eat regular meals in order to stabilize blood sugar levels. Habits like smoking, alcoholism, and caffeine consumption should be avoided to prevent worsening the panic attacks. It is also important to obtain useful advice from support groups for the better management of panic attacks [52,53]. Seeking Cognitive Behavioral Therapy (CBT) and other types of counselling can often be beneficial for individuals suffering from panic attacks [53]. Dental students experiencing panic attacks should be provided with the necessary counseling sessions or psychiatric consultation within the dental school in order to overcome such scenarios.

## 5. Conclusions

In the present study, we found that (42.9%) of the study participants reported having experienced panic attack symptoms. Female participants showed a higher prevalence of panic attacks when compared to males. Third year students displayed greater episodes of panic attacks compared to their respective counterparts. The most reported symptom of a panic attack was rapid or pounding heartbeat followed by breathlessness, chest pain, and shaking or trembling. It was also noted that most of the participants encountered a panic attack for the first time after joining dental school. The high occurrence of panic attacks among dental students emphasizes the importance of providing support programs and implementing preventive measures to help students, particularly those who are most susceptible to higher levels of these psychological conditions. Dental students experiencing panic attacks should be provided with the necessary counseling sessions or psychiatric consultation in order to overcome such scenarios. Dental schools should consider these findings when planning the dental curriculum; hence, the role of the faculty members is essential in these situations to provide support for the affected students.

## Figures and Tables

**Table 1 healthcare-11-02971-t001:** Demographic details of study participants.

Parameter	Sub-Parameter	Number	Percentage
Gender	Female	143	36.29%
	Male	251	63.71%
Academic Year	1st year	45	11.4%
	2nd year	47	11.93%
	3rd year	48	12.18%
	4th year	84	21.32%
	5th year	102	25.89%
	Intern	68	17.26%
University	Dar Al Uloom University	50	12.69%
	King Saud University	106	26.90%
	Princess Nourah Bint Abdulrahman University	64	16.24%
	Riyadh Elm University	64	16.24%
	Vision College	5	1.27%

**Table 2 healthcare-11-02971-t002:** Prevalence and symptoms of panic attacks among study participants.

Sl. No	Questions	Responses	Frequency	Percentage
1	Have you ever had any of the symptoms below? (Rapid or pounding heartbeat, Sweating, Trembling or shaking, Breathlessness, Chest pain or discomfort, Nausea, Dizziness or faintness, Feelings of unreality, Chills or hot flushes, Fear of losing control or going crazy, Fear of dying, Numbness or tingling, Feeling of choking) We defined a panic attack as a sudden rush of fear or discomfort accompanied by at least 4 of the symptoms listed below.	Yes	169	42.9
		No	225	57.1
2	When do you most experience panic attacks during the academic clinical year?	At the beginning of the year	22	13.02
		At the end of the year	38	22.49
		At the middle of the year	49	28.99
		Not determined	60	35.50
3	How many panic attacks did you encounter in the past four weeks?	0	58	34.32
		1–5	99	58.58
		6–10	12	7.10
4	When a panic attack occurs, for how long does it last?	less than 10 min	93	55.03
		10–30 min	57	33.73
		30–60 min	12	7.10
		more than one hour	7	4.14
5	For your panic attack, please rate on the below listed scale how anxious or worried you are of having another attack in the future.	No anxiety	41	24.26
		Mild anxiety	64	37.87
		Moderate anxiety	46	27.22
		Severe anxiety	18	10.65
6	Which of the following is the most common symptom you experience during panic attacks?	Rapid or pounding heartbeat	62	36.69
		Sweating	7	4.14
		Trembling or shaking	10	5.92
		Breathlessness	26	15.38
		Chest pain or discomfort	15	8.88
		Nausea	9	5.33
		Dizziness or faintness	7	4.14
		Feelings of unreality	9	5.33
		Chills or hot flushes	8	4.73
		Fear of losing control or going crazy	9	5.33
		Fear of dying	4	2.37
		Numbness or tingling	1	0.59
		Feeling of choking	2	1.18
7	When was the first time you experienced panic attack?	Before dental school	62	36.69
		After dental school	107	63.31

**Table 3 healthcare-11-02971-t003:** Triggering factors of panic attacks.

Sl. No	Questions	Responses	Frequency	Percentage
1	Please indicate in which of the following situation panic attacks have occurred?	Before or during exams, while giving a presentation, in the clinic while performing procedures, while under a lot of stress	88	52.07
		Before or during exams, in the clinic while performing procedures, while under a lot of stress	25	14.8
		Before or during exams, in the classroom during lectures, while giving a presentation, while under a lot of stress	19	11.2
		Before or during exams, in the clinic while performing procedures	16	9.4
		Before or during exams, while under a lot of stress	12	7.1
		Attacks occurred unexpectedly and “out of the blue”	9	5.3
2	Did you ever experience panic attacks outside of college?	Yes	115	68.05
		No	54	31.95

**Table 4 healthcare-11-02971-t004:** Impact of panic attacks among study participants.

Sl. No	Questions	Responses	Frequency	Percentage
1	Did your panic attack have an impact on your academic and/or clinical performance?	No impact (0%)	32	18.93
		Mild impact (10–30%)	80	47.34
		Moderate impact (40–60%)	48	28.40
		Severe impact (70–100%)	9	5.33
2	Do you think panic attacks are in some way harmful to your physical or mental health?	Yes	143	84.62
		No	26	15.38

**Table 5 healthcare-11-02971-t005:** Response of panic attacks among study participants.

Sl. No	Questions	Responses	Frequency	Percentage
1	After experiencing panic attack, were there any places you avoided or felt afraid of because of fear of having a panic attack?	None: no fear or avoidance	89	52.66
		Mild: occasional fear and/or avoidance	19	34.32
		Moderate: noticeable fear and/or avoidance	27	11.24
		Severe: extensive avoidance	3	1.78
2	After experiencing panic attack, were there any activities that you avoided or felt afraid of because they caused physical sensations like those you feel during panic attacks or that you were afraid might trigger a panic attack?	None: no fear or avoidance	72	42.60
		Mild: occasional fear and/or avoidance	63	37.28
		Moderate: noticeable fear and/or avoidance	27	15.98
		Severe: extensive avoidance	7	4.14
3	Have you ever consulted a counsellor to manage your panic attacks?	Yes	44	26.04
		No	125	73.96

**Table 6 healthcare-11-02971-t006:** Association of the occurrence of panic attacks in relation to gender and academic year.

Have You Ever Had Any of the Symptoms Below? (Rapid or Pounding Heartbeat, Sweating, Trembling or Shaking, Breathlessness, Chest Pain or Discomfort, Nausea, Dizziness or Faintness, Feelings of Unreality, Chills or Hot Flushes, Fear of Losing Control or Going Crazy Fear of Dying, Numbness or Tingling Feeling of Choking)						
Variables	Response	Frequency (n)	Percentage (%)	*p* Value	Chi Square Value	df
Gender						
Male	Yes	108	24.5	<0.001 *		1
	No	35	75.5		29.97	
Female	Yes	134	53.4			
	No	117	46.6			
Academic Year						
1st Year	Yes	16	35.6	0.01 *		
	No	29	64.4			
2nd Year	Yes	27	57.4			
	No	20	42.6			
3rd Year	Yes	28	58.3			
	No	20	41.7			
4th Year	Yes	36	42.9		14.53	5
	No	48	57.1			
5th Year	Yes	33	32.4			
	No	69	67.6			
Interns	Yes	29	42.6			
	No	39	57.4			
Enrolled University						
Dar Al Uloom University	Yes	12	24	<0.001 *		
	No	38	76			
King Saud bin Abdulaziz University for Health Sciences	Yes	58	55.2			
	No	47	44.8			
King Saud University	Yes	50	47.2			
	No	56	52.8		42.205	5
Princess Nourah Bint Abdulrahman University	Yes	38	59.4			
	No	26	40.6			
Riyadh Elm University	Yes	10	15.6			
	No	54	84.4			
Vision College	Yes	1	20			
	No	4	80			

Footnotes: * = significant.

## Data Availability

Data are contained within the article.

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
