# Peer review of "Prevalence, Symptoms, and Triggering Factors of Panic Attacks among Dental Students in Riyadh Saudi Arabia—A Cross Sectional Survey"

_healthcare, 2023, doi:10.3390/healthcare11222971_

Round 1

Reviewer 1 Report

Comments and Suggestions for Authors

Dear Authors,

Thank you for the opportunity to review the manuscript entitled "Prevalence, Symptoms and Triggering Factors of Panic Attacks Among Dental Students in Riyadh Saudi Arabia – A Cross Sectional Survey”.

The authors raise very important research issues from a practical and theoretical point of view regarding panic attacks among future dentists who will have to deal with patient anxiety in their professional work.  In my opinion, the level of scientific soundness and the overall merit of this manuscript is high. The article has many strengths.

The theoretical introduction is very well written, with references to the relevant literature.

The methods and procedures are very well described.

The results are presented in a very transparent, clear and consistent way. 

The discussion was also conducted at a high level, referring to comparisons of the authors' results with previous research results.

I believe that this article deserves to be published in Healthcare. I have only one small suggestion to improve the article.

I suggest that the authors of the article include in the results (describing differences or relationships) the values of the statistical tests obtained, e.g. the value of chi square, df, and p should be provided, and not just the p-value, as the authors did. Presenting the values in the Results section will certainly enhance the scientific character of this manuscript.

Best regards,

Reviewer

Author Response

Dear Reviewer,
Greetings!

Firstly I would like to thank you for your kind words of appreciation and encouragements.

I Sincerely thank you for your valuable inputs, and I would like to inform you that we have considered all the valuable comment suggested by you and have modified the manuscript as per your suggestions.

We have modified the manuscript to best of our knowledge, kindly consider the same and oblige,

Thank you and regards

Sincerely yours,

Dr. Sanjeev B Khanagar (Corresponding Author)

Reviewer 1

Comment 1: The theoretical introduction is very well written, with references to the relevant literature.

The methods and procedures are very well described.

The results are presented in a very transparent, clear and consistent way. 

The discussion was also conducted at a high level, referring to comparisons of the authors' results with previous research results.

I believe that this article deserves to be published in Healthcare. I have only one small suggestion to improve the article.

I suggest that the authors of the article include in the results (describing differences or relationships) the values of the statistical tests obtained, e.g. the value of chi square, df, and p should be provided, and not just the p-value, as the authors did. Presenting the values in the Results section will certainly enhance the scientific character of this manuscript.

Response: Thank you for valuable suggestion, please note that we have added the details of the of chi square, df, and p as per your suggestion.

Reviewer 2 Report

Comments and Suggestions for Authors

Dear authors,

I would like to appreciate all the efforts for this manuscript. There will be needed some corrections which I mentioned in the PDF file in the sticky note before considering for publication. Kindly find the attached pdf file.

Best of luck!

Comments on the Quality of English Language

Minor English corrections are required along with the proper punctuations.

Author Response

Dear Reviewer,
Greetings of the day!

Firstly I would like to thank you for kind words of appreciation and encouragements. I sincerely thank you for your valuable inputs, and would like to inform you that we have considered your valuable comments and have modified the manuscript as per your suggestions.

We are also providing point to point clarifications for the comments suggested by you. "Please see the attachment." PDF with responses to your valuable comments.

We have modified the manuscript to best of our knowledge, kindly consider the same and oblige,

Thank you and regards

Sincerely yours,

Dr. Sanjeev B Khanagar (Corresponding Author)

Reviewer 2

Comment: I would like to appreciate all the efforts for this manuscript. There will be needed some corrections which I mentioned in the PDF file in the sticky note before considering for publication. Kindly find the attached pdf file.

Response: Thank you for your valuable suggestion and inputs, please note that we have now added more details in manuscript as per your valuable suggestions. Please note that we have added our responses in the PDF along with the changes suggested by you.

Reviewer 3 Report

Comments and Suggestions for Authors

Thank you for the opportunity to review this article. The aim of the study is very interesting, but unfortunately I do not find the article suitable to be published in the current form. 

Firstly, it is difficult in the introduction to see the "red thread" in the text. The text would begin with more on mental health in general and then on mental health among students as arguments for this study. It is not clear why the gender of dental students is of interest. 

Secondly, the discussion requires a major revision. Now it consists just of a repetition of the results and no discussions of the results. There is further need of more references to articles on how to prevent panic attacks and to support students who suffer from panic attacks. The role of preventive measures and support is mentioned in the conclusion but have to be discussed.

Due to these problems with the article, I recommend major revisions.  

Comments on the Quality of English Language

Minor editing of English language required.

Author Response

Dear Reviewer,
Greetings of the day!

Firstly I would like to thank you for your valuable inputs, and would like to inform you that we have considered your valuable comments and have modified the manuscript as per your suggestions.

We are also providing point to point clarifications for the comments suggested by you. "Please see the attachment."

We have modified the manuscript to best of our knowledge, kindly consider the same and oblige,

Thank you and regards

Sincerely yours,

Dr. Sanjeev B Khanagar (Corresponding Author)

Reviewer 3

Comment: Firstly, it is difficult in the introduction to see the "red thread" in the text. The text would begin with more on mental health in general and then on mental health among students as arguments for this study. It is not clear why the gender of dental students is of interest. 

Response: We appreciate your concern, however we have followed a seminar approach in preparing this introduction section (general to specific). We have considered gender as a concern because the previous literature reports on higher amount of stress, anxiety and panic attacks among female participants. We have added more details on this aspect in the discussion section as per your valuable suggestion.

Comment: Secondly, the discussion requires a major revision. Now it consists just of a repetition of the results and no discussions of the results. There is further need of more references to articles on how to prevent panic attacks and to support students who suffer from panic attacks. The role of preventive measures and support is mentioned in the conclusion but have to be discussed.

Response: We thank you for your valuable suggestions, we have added more details in the discussion section and updated the references. We have also added an additional paragraph on the preventive and supportive measures in managing panic attacks in the discussion section as per your valuable suggestion.

Round 2

Reviewer 3 Report

Comments and Suggestions for Authors

I think that the manuscript has been sufficiently improved to be published in Healthcare.